# Environmental Factors in the Rehabilitation Framework: Role of the One Health Approach to Improve the Complex Management of Disability

**DOI:** 10.3390/ijerph192215186

**Published:** 2022-11-17

**Authors:** Lorenzo Lippi, Alessandro de Sire, Arianna Folli, Alessio Turco, Stefano Moalli, Antonio Ammendolia, Antonio Maconi, Marco Invernizzi

**Affiliations:** 1Physical and Rehabilitative Medicine, Department of Health Sciences, University of Eastern Piedmont “A. Avogadro”, 28100 Novara, Italy; 2Dipartimento Attività Integrate Ricerca e Innovazione (DAIRI), Translational Medicine, Azienda Ospedaliera SS. Antonio e Biagio e Cesare Arrigo, 15121 Alessandria, Italy; 3Physical and Rehabilitative Medicine Unit, Department of Medical and Surgical Sciences, University of Catanzaro “Magna Graecia”, Viale Europa, 88100 Catanzaro, Italy; 4SC Infrastruttura Ricerca Formazione Innovazione, Dipartimento Attività Integrate Ricerca Innovazione, Azienda Ospedaliera SS. Antonio e Biagio e Cesare Arrigo, 15121 Alessandria, Italy

**Keywords:** environment, environmental factors, disability, One Health, rehabilitation

## Abstract

Environment factors play a crucial implication in human health, with strong evidence suggesting that several biological, chemical, physical and social factors could be possible targets to implement effective strategies for human health promotion. On the other hand, a large gap of knowledge still exists about the implications of environmental factors in terms of functional impairment and disability, while the integration of an environmental-based approach in the therapeutic care of patients affected by disabilities remains still challenging. In this scenario, the One Health approach has been recently introduced in clinical care and aims to optimize health outcomes by recognizing the interconnection between people and the environment. Concurrently, the “Rehabilitation 2030 Initiative” proposed in 2017 by the WHO emphasized the need to integrate environmental-based strategies to promote rehabilitation across different health systems and different nations. However, no previous study underlined the potential implications of the One Health approach in the rehabilitation setting, nor the role of a comprehensive rehabilitation approach focused on environmental factors. Therefore, the aim of this narrative review was to present a comprehensive overview of the data currently available assessing the close relationship between rehabilitation and the environment to provide a different perspective on the comprehensive care of patients affected by disability.

## 1. Introduction

Disability is a global burden rapidly increasing worldwide with the age of the population, as reported by the Global Burden of Disease Study 2019 [1]. Concurrently, health expenditure related to disability is rising, emphasizing the need to develop sustainable strategies and policies to prevent and reduce the incidence of functional impairment [1]. In 2012, the World Health Organization (WHO) estimated that approximately 12.6 million deaths might be directly connected to environmental causes, consisting of 23% (95% CI: 13–34%) of the global causes of death worldwide [2]. On the other hand, it has been estimated that 22% (95% CI: 13–32%) of all disability-adjusted life years (DALY) were related to environmental factors [2]. Altogether, these data emphasized that environmental factors play a key role in global health, with crucial consequences in terms of social and sanitary costs [2]. In this scenario, environmental health research defined the environment as a set of biological, chemical, and physical factors coupled with social aspects that together constitute hazards or risks to human health [3]. Moreover, it should be noted that sex, ethnicity, socioeconomic status and geographical location could be considered the main social determinants of health and strongly influence access to health services [4]. In this context, recent research has proposed a novel approach defined as One Health, a collaborative, multi-sectoral, and transdisciplinary approach that takes into account the interconnectedness between humans, animals, plants, and the environment. It is aimed at achieving optimal health outcomes by taking into consideration the interaction between humans and the environment [5]. Environmental factors might represent one of the most important barriers for patients with disability [6]; there is still a gap of knowledge in the current literature about the integration of the One Health approach in the therapeutic management of patients affected by disabilities.

Physical and rehabilitation medicine is a branch of medicine focusing on functional improvement and enhancing the quality of life of patients affected by disability through specific rehabilitation programs and interventions [7]. However, an individualized rehabilitation plan should take into account several environmental factors affecting physical function, independence of activity of daily living, social participation, psychological well-being and crucially affecting the functional recovery of patients with disability [8]. Despite these considerations, to the best of our knowledge, no previous study underlined the importance of sustainable strategies targeting environmental factors in the rehabilitative treatment framework of disability.

Therefore, the objective of this narrative review was to characterize the strict linking between environment and disability in order to provide a different point of view about the comprehensive rehabilitation management of environmental disorders and integrate the One Health approach in the tailored management of patients with disability.

## 2. Research Methodology

This narrative review has been realized following the SANRA quality criteria [9]. Scientific literature research has been performed on PubMed/Medline, Web of Science (WoS), and Scopus using the following Mesh terms: “Environmental risk factors”, “Environment”, “Environmental Medicine”, “Disability”, “Function”, “Impairment”, and “Rehabilitation”. Table 1 summarizes the SPIDER tool search strategy [10].

The literature research was performed between June 2022 and September 2022 by two independent reviewers (L.L. and A.d.S.). Subsequently, two reviewers (L.L. and A.d.S.) independently screened the studies for eligibility. If a consensus was not reached, a third reviewer was asked (M.I.).

We considered eligible all the articles answering the research question: “does the environment affect the risk of disability?”.

More in detail, inclusion criteria were: (i) studies addressing human subjects; (ii) studies addressing environmental disorders or environmental factors affecting human health; (iii) studies assessing disability, physical function, and Health-related Quality of Life (HR-QoL). Exclusion criteria were: (i) all studies in languages other than English; (ii) studies without full text available; (iii) studies involving animals; (iv) conference abstracts, masters, or doctorate theses.

A qualitative method has been used for data extraction and data synthesis. More in detail, environmental disease characteristics, the World Health Organization’s position, environment-related disability, and the One Health approach were extracted and synthesized in the manuscript. Both the data extraction and the data synthesis have been independently performed by two reviewers (L.L. and A.d.S). In case of disagreement, a third reviewer was asked (M.I.). Due to the heterogeneity of the studies considered and in accordance with the narrative review design, a qualitative synthesis has been performed and all outcome data were reported in a narrative way.

## 3. Environmental Diseases

The World Health Organization (WHO) defines environmental health by referring to the chemical, physical and biological factors external to a person, their choices and behaviors [11]. Several environmental agents have crucial implications for human health [12]. For instance, pollution of water, air, chemical or biological agents might play a pivotal role in the pathological pathways involved in environmental disease development [13]. Moreover, UV rays and ionizing radiation, noise, electromagnetic fields, and occupational risks might be considered environmental factors significantly affecting human health. Interestingly, behaviors connected to environmental factors (e.g., availability of safe water to be able to wash hands or physical activity promoted by urban planning aimed at contrasting the sedentary lifestyle) are considered factors significantly involved in environmental diseases development [14]. On the contrary, the consumption of alcohol, tobacco, and substances of abuse, are not considered related to environmental disorders, given that environmental modifications do not affect the risk of disease [11]. Some examples are: (a) The pathologies due to passive smoke are related to the environment, but not pathologies in active smokers; (b) malnutrition in less developed countries is considered an environmental pathology, on the contrary obesity due to bad eating habits is not considered a result of environment interaction; (c) trauma due to road accidents (pathologies due to mechanical energy transmission) are generally considered environmental pathologies, as the intervention in the environment (roads that are intact and better structured for traffic) leads to their reduction [2].

In general, environmental pathologies are multifactorial diseases that develop as the result of interactions with multiple risk factors, both environmental and non-environmental related [13]. Furthermore, the risk associated with specific environmental factors is probabilistic (exposure has not been associated with certain diseases) and increases with the intensity and duration of exposure [2]. In particular, in the pediatric age, there is a significant environmental contribution to the development of several infectious diseases, neonatal and nutritional pathologies and traumas [15]. On the other hand, in the elderly, chronic pathologies related to environmental factors show a higher prevalence, while traumatic injuries still remain important factors affected by the environment [16].

In this scenario, preventive strategies aimed at creating healthier environments might have a crucial role in most disease control strategies [17]. For instance, about 20% of cancer cases are globally associated with modifiable environmental factors and the current preventive strategies mainly focus on atmospheric pollution radiation reduction, the protection of exposed workers and the correct management of chemical agents [11].

Concurrently, approximately 35% of lower respiratory tract infections might be related to modifiable environmental factors [11]. Therefore, home environment, atmospheric pollution, and passive smoking are potential targets for preventive strategies aiming at reducing the occurrence of the disease [2].

In this context, it is not surprising that there is growing attention on international policies on environmental diseases [2]. In particular, the WHO is currently supporting several initiatives aimed at creating healthier environments and healthier populations, including: (1) to promote good governance in the areas of health and environment while guiding significant transformations in other areas such as transportation and energy; (2) to implement research focusing on monitoring changes in health hazards in order to improve solutions and information and promote evidence-based standards and effective remedies; (3) assisting nations with methods for expanding their actions and capacity building; (4) to enhance emergency planning and response capabilities in the event of environmental events and offer pertinent advice on environmental health services and workplace health and safety [18]. Moreover, in the Compendium of WHO and other UN guidance on health and environment, some countermeasures were proposed for both people and organizations to reduce environmental diseases worldwide [11]. In this context, these countermeasures deal with clean air, a constant climate, enough water, sanitation and hygiene, safe chemical usage, radiation protection, healthy and safe workplaces, sound agricultural practices, health-supportive towns and built environments and protected nature [2].

Altogether, these data emphasized that environmental diseases are a crucial issue in the current literature, with growing attention to the international policies targeting environmental modification to improve global health [2]. In this scenario, several environmental factors might be associated with human health and might be effectively targeted by precise environmental strategies [19]. However, environmental modifications might have a crucial impact not only on global health but also on functional impairment. Interestingly, recent research proposed that environmental modification might be considered in the rehabilitation research setting in order to improve the physical function of patients affected by disability [20,21].

## 4. Functional Impairment and Environment

### 4.1. Does the Environment Affect the Risk of Disability?

The environment plays a crucial role in the development of disabling conditions, with growing research underlining that a comprehensive rehabilitation approach should include environmental factors to optimize the functional recovery and social integration of patients with disability [22]. In addition, it has been previously demonstrated how the environment could positively or negatively affect psycho-physical well-being. In recent years, attention has been paid not only to the psycho-physical sphere but also to the social sphere, with a growing interest in a patient-centered approach to improve not only physical health but also social participation and HR-QoL [23].

In this scenario, a holistic approach to managing disability should take into account several environmental factors that might promote disability. Table 2 better characterized the evidence supporting the effects of environmental factors on the risk of disability.

### 4.2. Environment and Healthy Aging, Frailty and Disability

In the past decade, growing attention has been raised to healthy aging due to the progressive aging of the population worldwide [48]. In this context, the WHO, Member States and Partners for Sustainable Development Goals proposed a Global Strategy and Action Plan for Ageing and Health for 2016–2020 and the WHO program The Decade of Healthy Ageing 2020–2030 [48].

In older adults, environmental factors play a crucial role in maintaining or promoting a healthy lifestyle with significant implications for functional capacity and the risk of disability [49,50]. On the other hand, several risk factors related to the environment might promote frailty, functional disability, cognitive impairment, hospitalization and mortality in older adults [51]. However, in the current literature, there is still a large gap of knowledge about the possible link between environmental risk factors and disability, with most studies on this topic focusing on the relation between a specific exposure and a particular disease, without focusing on a direct link between functional impairment and environmental factors. Interestingly, the recent study by Yu et al. [52] assessed the role of environmental factors and physical frailty, reporting that living in neighborhoods with a higher percentage of green space directly reduces the risk of frailty. Similarly, Lee et al. [53] subsequently underlined that PM exposure might be significantly associated with prefrail and frailty status in elderly subjects. On the other hand, a significant association between psychological distress and a poor social network has been reported as an important risk factor for frailty syndrome in elderly subjects [54].

Altogether, these studies underlined that environmental factors should be considered together with lifestyle and psychosocial factors to develop effective strategies aimed at reducing the risk of frailty in older adults. However, a comprehensive approach to frailty syndrome should include a tailored approach to functional impairment frequently characterizing this condition. In this context, the recent study by Momosaki et al. [55] underlined that malnutrition and nutritional opportunities play a crucial role in modulating physical function, in line with the “rehabilitation nutrition approach” to a disability, and a comprehensive synergistic intervention including nutritional support and physical exercise.

In addition, Chen et al. [56] showed that long-term exposure to NO_2_ is associated with an accelerated decline in static lung volume, and diffusion capacity. These modifications could be a risk factor for restrictive lung disorders in the elderly, with significant implications for physical performance. Concurrently, the study by Cassou et al. [57] highlighted a close relationship between exposure during working life to specific occupational risk factors (such as noise, heat, dust, carrying heavy loads, and awkward postures on the one hand) and physical disability after retirement.

While occupational risk factors might affect functional performance, climatic changes might significantly interact with physical activity levels with crucial implications for physical performance in the elderly [58]. In addition, the study by Zanobetti et al. [59] underlined a greater risk of cardiovascular dysfunction at lower and higher temperatures in older men, with an increased risk of ventricular arrhythmias. Although physical function plays a pivotal role in activity of daily living (ADL) independence and the need for assistance in older adults, cognitive function impairment is a common condition in these patients and represents a global burden for both sanitary and assistance costs. In this context, several risk factors have been identified to have a role in cognitive impairment development in older adults. More in detail, Dardiotis et al. [60] suggested that people living in areas near sprayed fields (exposed to pesticides) had poorer neuropsychological performance in language, executive and visual-spatial functioning than those who had never lived in these areas. Moreover, Wueve et al. [61] showed that an increment of 10 A-weighted decibels in noise corresponds to 36% and 29% higher odds of prevalent Mild Cognitive Impairment (MCI) and Alzheimer’s disease. Similarly, in 2022 Gao et al. [62] demonstrated that long-term ozone exposure increases the risk of cognitive impairment with a linear trend. Therefore, several environmental exposures might significantly affect cognitive function and should be considered in a tailored approach aimed at maximizing functional independence in older adults.

### 4.3. Sanitary Costs, Opportunities, and Barriers to Independence

As previously reported, environmental factors might have detrimental consequences on public health, with a significant impact in terms of sanitary, social and assistance costs [11].

Therefore, environmental factors have been considered a potential target to reduce health care expenditure in the future years, with growing interest in organizational campaigns and social strategies aimed at reducing the negative environmental implications on global health [63]. For instance, the study by Lightwood et al. [64] reported that passive smoking elimination would immediately prevent $1.5 to $2.3 billion in costs annually for coronary heart disease treatment over the next 30 years.

The prevalence of smoking has decreased significantly since 1990 with significant positive effects on rates of heart, stroke and cancer-related deaths, and crucial implications in sanitary expenditure and DALY related to environmental factors [65].

In this scenario, close surveillance, research, multilevel interventions, environmental modifications and strong health policies represent effective therapeutic options to improve global health and significantly reduce sanitary and assistance costs related to the disabling consequences of environmental diseases [66]. However, evidence-based practices and interventions are needed for an ecologically comprehensive approach to changing environmental determinants and capitalizing on the concept of reciprocal determinism [67]. Therefore, the environmental modifications might be integrated into several settings, including rehabilitation. Attention should be paid not only to the prevention of disabling conditions but also to the barriers limiting patients’ engagement in rehabilitation programs and barriers limiting the social participation of patients with disabilities that might need environmental adaptation to optimize their ADL independence.

## 5. One Health Approach

The One Health approach has been defined by the Centers for Disease Control and Prevention (CDC) as “a collaborative, multisectoral, and transdisciplinary approach—working at the local, regional, national, and global levels—aiming at achieving optimal health outcomes recognizing the interconnection between people, animals, plants, and their shared environment” [5]. Despite the One Health approach having been developed as early as the XIX century, it has been rediscovered in the past few years as highlighted by growing literature focusing on this cutting-edge approach [5]. Interestingly, One Health emphasizes the concept that health is a fundamental right that must be shared between humankind, the environment, and animals. Moreover, it is suggested that human health is closely linked to the health of the fauna and environment. Thus, these three macro-areas share several influencing factors, and their dynamics and interactions are constantly evolving: the growing human population, the evolution of housing habits, climate change, the extensive and often reckless use of the land, the ease of movement of living beings from one part of the globe to another. Taken together all these factors have drastically affected human diseases [68].

From a comprehensive point of view, the One Health approach perfectly overlaps with the aim of the United Nations 2030 Agenda for Sustainable Development Goals (SDGs), targeting peace and prosperity for Earth’s inhabitants and their environment [69]. One Health aims at controlling and limiting zoonoses (which according to some estimates represent over 60% of known human pathogens and 75% of emerging pathogens [69]), antibiotic resistance, food safety, environmental contamination and, in general, the health-related risks shared by people, animals, and the environment. Furthermore, the One Health approach promotes its multidisciplinary method in the comprehensive management of chronic diseases, non-communicable diseases, mental health, work accidents and occupational health [5].

In this scenario, the recent study by Sterckx et al. [70] has proposed a One Health-based protocol for the holistic management of burnout syndrome. Interestingly, the authors emphasized the strict relationship between mind, body, and emotions within the individuals, integrating the patient in a broader context, including both social and environmental factors.

In this scenario, growing efforts have been recently paid to digital innovation and data sharing; realizing shared biobanks aimed at reorienting not only the target to human health but also to animals and the environment. Remarkably, the data present in these biobanks could have a role in promoting interconnections between different areas and assessing the multilevel interactions among the potential environmental factors in human pathologies [71]. This novelty approach might have a role in developing future prospective translational studies, enhancing the clinical application of laboratory data into the clinical setting and from the clinical settings in organizational models, in order to optimize sustainable strategies to manage human health with a One Health approach [72].

In this context, a strict connection might associate the One Health approach with rehabilitation. More in detail, rehabilitation is historically characterized by multidisciplinary and interdisciplinarity features aimed at optimizing the functional recovery of people with disability, enhancing the interaction with the environment and developing effective strategies and environmental modifications to promote independence in ADL, psychological well-being and/or efficient social-work-familiar integration.

In conclusion, the One Health approach could be effectively integrated into the complex framework of disability management paving the way to a broader concept of human health. Therefore, One Health is a novel comprehensive rehabilitation approach that takes into account not only the patient’s individuality but also the environmental factors.

## 6. One Health Approach and Disability

Physical and Rehabilitation Medicine (PRM) is a branch of medicine that aims at promoting functional recovery in patients affected by disability [73,74]. Due to its intrinsic characteristics, PRM should not focus on a single apparatus but should contemplate a holistic view of the functional integrity of the whole patient [75].

Despite disability onset being strongly associated with environmental pathologies, several environmental factors might have a crucial role in the comprehensive treatment of disability in both post-acute and chronic settings [76,77]. More in detail, environmental factors that should be considered for a comprehensive rehabilitation plan include the socio-residential context on the one hand and, on the other, the territory in which the patient lives as well as the accessibility to rehabilitation care in both outpatient and community settings [76,77].

The social environment crucially affects patient opportunities to perform rehabilitation programs, due to the heterogeneity in the rehabilitation settings and the scarce healthcare worker knowledge about optimal rehabilitation interventions [78]. Furthermore, home environment and caregiver engagement play a key role in rehabilitation programs, influencing the home care management of patients with disability and improving the translation of functional improvement from the rehabilitation setting to the activity of daily living [79]. Interestingly, technological advances and digital innovation have been raised to promote access to healthcare services and rehabilitation and have grown in attention in recent years. During the COVID-19 pandemic, these approaches spread in clinical contexts as a result of the modifications of environmental factors requiring reduced contact between patients and healthcare professionals [80]. In this scenario, telemedicine and telerehabilitation could provide healthcare to patients affected by different diseases even from remote areas [81,82,83].

Lastly, the work environment might be important to support the return to work after a rehabilitation program, and the literature suggests that a supportive, adapted and protected work environment allows for psychological well-being and economical independence in patients undergoing a rehabilitation program [84].

Altogether, these data emphasize that environment might have a crucial impact on rehabilitation adherence, rehabilitation results, and rehabilitation translation in the activity of daily living, with significant implications for patient quality of life. Figure 1 summarizes the One Health approach to disability.

Lastly, the One Health approach in the rehabilitation field is in line with the “Rehabilitation 2030 Initiative” [85], proposed in 2017 by WHO. Interestingly, this document identified ten areas of priority work to strengthen rehabilitation across health systems and across nations [85]. More in detail, the environment is a key target in one of these areas, since “Building comprehensive rehabilitation service delivery models to progressively achieve equitable access to quality services, including assistive products, for all the population, including those in rural and remote areas” [85].

In conclusion, there are several suggestions on how the One Health approach might be integrated into the complex rehabilitation framework to treat patients affected by disability. This could pave the way to implement interdisciplinary teamwork that might overcome barriers to rehabilitation accessibility and cover patients’ needs for rehabilitation. Therefore, future research might focus on the integration of the One Health concept in rehabilitation settings to implement the effectiveness of the complex management of disability.

## 7. Conclusions

Environmental factors have been considered determinants of several pathological conditions and might also affect the risk of disability. However, to date, there is still a large gap in knowledge regarding the specific interaction between environmental factors and disability. In this context, the integration of environmental modifications in the complex rehabilitation framework might be considered to improve independence in ADL and reduce assistance costs, especially considering the progressive aging of the population and the increase in age-related disorders.

In this scenario, the One Health approach might be considered a suitable option to integrate environmental factors into the comprehensive management of chronic disabling diseases. This could help to overcome barriers to patients’ engagement and enhance access to rehabilitation services, and implement the rehabilitation framework to create more effective and sustainable strategies to counter disability.

## Figures and Tables

**Figure 1 ijerph-19-15186-f001:**
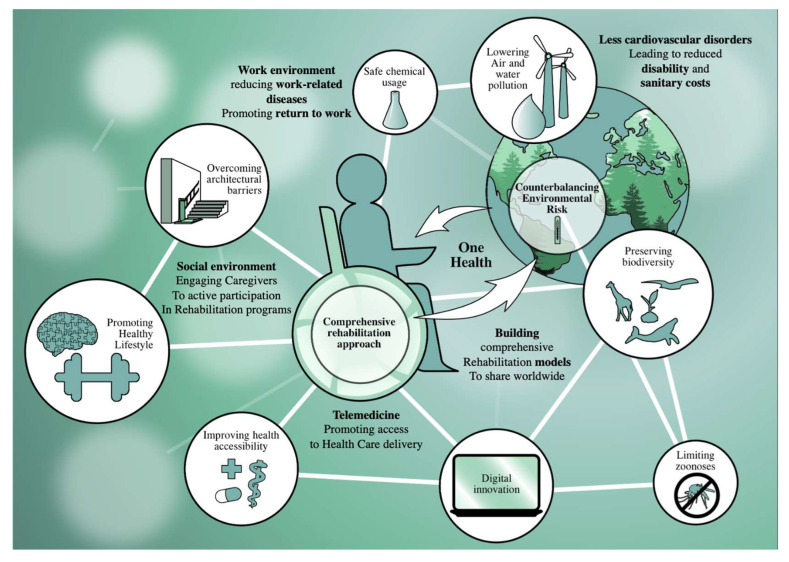
One health approach integrated into the modifiable environmental factors affecting disability.

**Table 1 ijerph-19-15186-t001:** Spider tool search strategy.

S	PI	D	E	R
Sample	Phenomenon of Interest	Design	Evaluation	Research Type
Human subjects	Environment	Any	Disability	Qualitative
	“Environmental risk factors” “Environment” “Environmental Medicine”		“Disability” “Function” “Impairment” “Rehabilitation”	

**Table 2 ijerph-19-15186-t002:** The effects of environmental factors on risk of disability.

Environmental Factors	Effects on Human Health and Disability
**Environmental** **pollution**	Environmental pollution represents one of the most studied environmental factors leading to environmental diseases, with approximately 9 million premature deaths every year and might be related to environmental pollution [24]. In this scenario, fine particulate matter < 2.5 µm (particulate matter—PM2.5) air pollution is one of the most important environmental risk factors leading to cardiovascular disorders commonly correlated with significant functional impairment, disability, high sanitary and assistance costs and even mortality [25,26,27].
**Noise pollution**	Chronic noise pollution plays an important role in the disability development process, with significant implications in the work setting. Beyond the widely documented effects of reduction in hearing sensitivity [28], noise pollution might be related also to stress, cardiovascular disease and cognitive impairment risk [28].
**Occupational risks**	Limitation of exposure to occupational risk factors and effective protection strategies have a crucial role in reducing the incidence of occupational diseases with relevant implications in terms of disability. More in detail, work-related musculoskeletal disorders are highly prevalent conditions related to work overuse affecting muscles, nerves, tendons, joints, cartilage, and spinal discs [29]. In this scenario, the functional limitation and pain symptoms might be effectively targeted by a comprehensive rehabilitation plan, which might include environmental and ergonomic modifications. On the other hand, vapors, gas, dust, or fumes might promote the development of pathological respiratory conditions (emphysema, chronic bronchitis, chronic obstructive pulmonary disease, silicosis, mesothelioma, lung cancer) leading to an increased incidence of functional impairment, reduced physical performance and higher need of assistance [30,31].
**Climatic change**	Climate changes might influence the onset of several health issues, including cardiovascular disorders [32], musculoskeletal painful conditions [33], psychiatric health status [34] and even death [35].
**Urban, suburban or** **rural setting and green areas**	In recent years, it has been observed that non-communicable diseases (NCDs), including diabetes, obesity, and heart disease, are strongly associated with urbanization [36], and urban or rural setting plays a key role in the prevalence of these environmental diseases. In this context, recent research suggested that rural and urban organization of the areas might affect not only the prevalence of environmental diseases but also their clinical course [37,38]. In addition, it has been reported a close link between exposure to residential green spaces and improved health outcomes in urban populations [39,40,41,42].
**Socioeconomic** **condition**	Socioeconomic condition significantly affects human health leading to disparities in accessibility to healthy behaviors and healthcare service. In particular, socioeconomic conditions appear to be related primarily to body composition, obesity and power physical activity levels [43]. According to the study by Feng et al. [44], the mean BMI was higher in the poorest socioeconomic condition, with increased inactivity time and poor physical activity levels. An interesting analysis of social control by Karriker-Jaffe et al. [45] reported that a family history of alcoholism might exacerbate a high-risk drinking trend. To decrease the high-risk of drinking and alcohol issues, policymakers should consider the differential advantages of limiting alcohol access for persons from high-risk households [46]. In addition, healthcare service deliveries are recently implemented with digital innovation solutions to boost the management of functional impairment. Telemedicine and telerehabilitation solutions are growing solutions that have been proposed to overcome barriers to healthcare accessibility [47]. However, it has been reported that social disparities might reduce accessibility to technological interventions for patients with socioeconomic disadvantages.

## Data Availability

Not applicable.

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
