# Peer review of "Environmental Factors in the Rehabilitation Framework: Role of the One Health Approach to Improve the Complex Management of Disability"

_ijerph, 2022, doi:10.3390/ijerph192215186_

Round 1

Reviewer 1 Report

ABSTRACT

The research topic is of scientific and social interest.

The topic is threat in correct form

In general, the article is very interesting, clarifies the purpose of the study and I consider that the topic is in line with the journal’s research objectives.

INTRODUCTION:

The study objective is well defined and identified in both the abstract and the introduction, well referenced. Clear the aims of the article

The structure is well defined, and it has enough information in separated  areas.

It’s been a good thing the references articles, the structure is well done, clear and focused.

One Health Approach

It’s been a good thing the references articles, the structure is well done, clear and focused.

One Health Approach and disability

It’s been a good thing the references articles, the structure is well done, clear and focused.  Very interesting point of view

DISCUSSION:

The conclusions and the discussion are well drawn and interesting. Appropriate extra notes in these parts.

An interesting line of research is observed

CONCLUSION:

An interesting line of research is observed, a bite short, but it´s correct.

The main question addressed by the research is The relation between environment and health.

I consider the topic is original, because it has an important investigation line.

Regarding the methodology, it colud be better with some more graphics. It colud be more information at the figure.

Author Response

Dear Reviewer,

thank you for your letter and kind comments concerning our manuscript. We would like to express our sincere appreciation for your careful reviewing and invaluable comments which help us to further improve this paper.

The revisions based on your comments have been highlighted in the manuscript in yellow, and our detailed responses according to each revision are shown as followed.

Regarding the methodology, it could be better with some more graphics.

We would like to thank the Reviewer for the insightful comment. We improved the methodology section including a Table for the search strategy in accordance with the Reviewer’s instructions.

It colud be more information at the figure.

We would like to thank the Reviewer for the insightful comment. We improved the Figure in accordance with the Reviewer’s instructions.

Reviewer 2 Report

1 This is a review work, easy to read.

2 The item is current and in line with the main concerns of the WHO.

3 Even though it was a topic for review, it suggested having a perfectly identified objective, considering the enormous diversity of contents in the area.

4 The methodology used in the eligibility of studies needs to be clarified. The inclusion or exclusion criteria of an article are not clearly identified. We don't know the reviewers' criteria, much less their training for this task.

5 We also do not know how the questions were constructed, for example "does de environment affect the risk of dissability'"...

6 It would also be desirable to be careful not to confuse quality of life with functionality.

7 The conclusion should be more objective and, in a way, answer the question of the review.

Author Response

Dear Reviewer,

thank you for your letter and kind comments concerning our manuscript. We would like to express our sincere appreciation for your careful reviewing and invaluable comments which help us to further improve this paper.

The revisions based on your comments have been highlighted in the manuscript in yellow, and our detailed responses according to each revision are shown as followed.

3 Even though it was a topic for review, it suggested having a perfectly identified objective, considering the enormous diversity of contents in the area.

We would like to thank the Reviewer for the insightful comment. We better characterized the objective of the study at the end of the introduction section, in accordance with the Reviewer’s instructions.

4 The methodology used in the eligibility of studies needs to be clarified. The inclusion or exclusion criteria of an article are not clearly identified. We don't know the reviewers' criteria, much less their training for this task.

We would like to thank the Reviewer for the insightful comment. We improved the methodology section in accordance with the Reviewer’s suggestion. Moreover, we characterized who performed the analysis; their experience in research areas and methodology is testified by the following review already published in international peer reviewed journals.

https://pubmed.ncbi.nlm.nih.gov/?term=Lippi+L&filter=pubt.review

https://pubmed.ncbi.nlm.nih.gov/?term=de+Sire+A&filter=pubt.review

https://pubmed.ncbi.nlm.nih.gov/?term=Invernizzi+M&filter=pubt.review

5 We also do not know how the questions were constructed, for example "does de environment affect the risk of dissability'"...

We would like to thank the Reviewer for the insightful comment allowing us to further improve the paper. The research question has been better characterized in methodology section in accordance with the Reviewer’s suggestions.

6 It would also be desirable to be careful not to confuse quality of life with functionality.

We would like to thank the Reviewer for the insightful comment. We carefully revised the whole article for this issue and the manuscript has been improved in accordance with the Reviewer’s suggestion.

7 The conclusion should be more objective and, in a way, answer the question of the review.

We would like to thank the Reviewer for the insightful comment. We improved the conclusion Section in accordance with the Reviewer’s instructions.

Reviewer 3 Report

  Thank you for the opportunity to review the manuscript entitled: Environmental factors in the rehabilitation framework: role of the One Health approach to improve the complex management of disability.

 This is an interesting topic with a highly quality and relevance for patients with disability and the public in general.  

However, I have some comments to the authors.

I believe the paper should describe the results of the literature review, and report the findings according with the guidelines for this type of search. It is clear that the paper is a narrative review but the authors performed a clear search and literature review.

Please review section 3. All the environmental risks are highlighted as “very important” and influencing rehabilitation programs so a table would be more concise and descriptive.

Additionally, the readers of this journal have different backgrounds so the One Health Approach should be mentioned at the beginning of the paper instead of being in number 5. It helps to understand the aims of the paper better.  

Please clarify the following acronyms, since this is the first time that are mentioned in the paper and as I mentioned, readers have different backgrounds.

Line 157: HR-QoL

Line 227: PM

Line: ADL

Line 10 A-weighted

Line 259: MCI

Also in line 100, Interaction is with capital letter. Please review.

Sentence in lines 260-263 is not clear. Please rewrite.

Line 362 should be “health care professionals”.

Congratulations for this interesting paper   

Author Response

Dear Reviewer,

thank you for your letter and kind comments concerning our manuscript. We would like to express our sincere appreciation for your careful reviewing and invaluable comments which help us to further improve this paper.

The revisions based on your comments have been highlighted in the manuscript in yellow, and our detailed responses according to each revision are shown as followed.

I believe the paper should describe the results of the literature review, and report the findings according with the guidelines for this type of search. It is clear that the paper is a narrative review but the authors performed a clear search and literature review.

We would like to thank the Reviewer for the insightful comment. We characterized the literature search in order to fully meet the SANRA quality criteria for narrative review. We better characterize this issue in the methods section. According to the SANRA criteria, a specific results section should not be provided. Morevoer, we better characterized the qualitative synthesis methodology of the study results.

Please review section 3. All the environmental risks are highlighted as “very important” and influencing rehabilitation programs so a table would be more concise and descriptive.

We would like to thank the Reviewer for the insightful comment. We provided a Table in accordance with the Reviewer’s instructions.

Additionally, the readers of this journal have different backgrounds so the One Health Approach should be mentioned at the beginning of the paper instead of being in number 5. It helps to understand the aims of the paper better.  

We would like to thank the Reviewer for the insightful comment. We characterized the One Health Approach in the background section in accordance with the Reviewer’s suggestion.

Please clarify the following acronyms, since this is the first time that are mentioned in the paper and as I mentioned, readers have different backgrounds.

Line 157: HR-QoL

Line 227: PM

Line: ADL

Line 259: MCI

We would like to thank the Reviewer for the comment. We clarified all the acronyms following the Reviewer’s suggestion.

Also in line 100, Interaction is with capital letter. Please review.

We would like to thank the Reviewer for the comment. We corrected the typing error.

Sentence in lines 260-263 is not clear. Please rewrite.

We would like to thank the Reviewer for the comment. We rewrote the sentence in accordance with the Reviewer’s suggestion.

Line 362 should be “health care professionals”.

We would like to thank the Reviewer for the comment. We corrected the typing error.

Congratulations for this interesting paper   

Thank you for your kind comments concerning our manuscript. We would like to express our sincere appreciation for your careful reviewing,
